

# Determining the Optimal Grid Resolution for Topographic Analysis on an Airborne Lidar Dataset

Taylor Smith[1], Aljoscha Rheinwalt[1], and Bodo Bookhagen[1]

[1]Institute of Earth and Environmental Sciences, Universität Potsdam, Germany

**Correspondence:** Taylor Smith (tasmith@uni-potsdam.de)

**Abstract.** Digital Elevation Models (DEMs) are a gridded representation of the surface of the earth and typically contain uncertainties due to data collection and processing. The topographic metrics slope and aspect contain errors and uncertainties inherited both from the representation of a continuous surface as a grid (referred to as truncation error, TE), and from any DEM uncertainty. We analyze in detail the impacts of TE and propagated elevation uncertainty (PEU) on slope and aspect.

Using synthetic data as a control, we define functions to quantify both TE and PEU for arbitrary grids. We then develop a quality metric which captures the combined impact of both TE and PEU on the calculation of topographic metrics. Our quality metric allows us to examine the spatial patterns of error and uncertainty in topographic metrics, and to compare calculations on DEMs of different sizes and accuracies.

Using lidar data with point density of ~10 pts/m$^2$ covering Santa Cruz Island in southern California, we are able to generate
DEMs and uncertainty estimates at several grid resolutions. Slope (aspect) errors on the one-meter dataset are on average 0.3° (0.9°) from TE, and 5.5° (14.5°) from PEU. We calculate an optimal DEM resolution for our SCI lidar dataset of four meters that minimizes the error bounds on topographic metric calculations due to the combined influence of TE and PEU for both slope and aspect calculations over the entire SCI. Average slope (aspect) errors from the four meter DEM are 0.25° (0.75°) from TE and 5° (12.5°) from PEU. While the smallest grid resolution possible from the high-density SCI lidar is not
necessarily optimal for calculating topographic metrics, high point-density data are essential for measuring DEM uncertainty across a range of resolutions.

## 1 Introduction

Continuous surfaces are often projected onto an evenly sampled grid – digital elevation models (DEMs) are a common example. The accuracy of the gridded representation of the underlying surface is controlled by the spacing of the grid, the variability of
the surface (i.e., the terrain itself), and the amount of uncertainty added during data collection and processing.

In recent years, an ever-growing array of DEM datasets have become available across a range of resolutions and spatial scales. As new data acquisition and processing strategies – such as lidar, stereo photogrammetry, and structure from motion – mature, the number and variability of features represented in a DEM will grow dramatically. In this manuscript, we refer to DEMs as high resolution when their grid spacing is low (e.g., a one meter DEM), and low resolution when their grid spacing
is high (e.g., a 30 meter DEM). Across all DEM resolutions, it is important to quantify the uncertainties in DEMs, and their

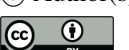



potentially large impacts on metrics calculated from the DEM (Florinsky, 1998; Zhou and Liu, 2004; Oksanen and Sarjakoski, 2005; Wechsler and Kroll, 2006; Wechsler, 2007).

The estimation of topographic metrics, such as slope and aspect, is a common task across scientific disciplines. Slope and aspect provide important boundary conditions for hillslope stability analysis (e.g., Montgomery and Dietrich, 1994; Tucker and Bras, 1998) and landslide monitoring (e.g., Guzzetti et al., 1999; Ayalew and Yamagishi, 2005), geomorphic transport laws (e.g., Roering et al., 1999; Dietrich et al., 2003; Pelletier, 2008; Grieve et al., 2016), hydrologic applications (e.g., Band, 1986; Zhang and Montgomery, 1994; Tarboton, 1997; Pelletier, 2010, 2013) including flood risk modeling (e.g., Ouma and Tateishi, 2014), tectono-geomorphic modeling (e.g., Whipple and Tucker, 1999; Snyder et al., 2000; Tucker and Hancock, 2010; Kirby and Whipple, 2012; Bookhagen and Strecker, 2012), ecologic modeling (e.g., Franklin, 1995; Guisan and Zimmermann, 2000; Thompson et al., 2001), and vegetation analysis (e.g., Pierce et al., 2005; Kent, 2011).

A wide range of methods have been developed to accurately derive slope and aspect from elevation data using a range of approaches, each optimized for different use cases (e.g., Evans, 1980; Zevenbergen and Thorne, 1987; Skidmore, 1989; Tarboton, 1997; Dunn and Hickey, 1998; Schmidt et al., 2003). For example, the methods D-8 and D-Infinity are optimized towards hydrologic flow-routing problems, particularly on DEMs of low spatial resolution (Tarboton, 1997), and the methods of Evans (1980) and Zevenbergen and Thorne (1987) include some smoothing of the underlying DEM to minimize the impacts of DEM noise. However, all of these methods contain some intrinsic error due to truncation error (TE) – the deviation of the gridded sample from the continuous original surface (Figure 1). The TE describes the error made by truncating an infinite sum and approximating it by a finite sum. It often includes a discretization error, which arises from taking a finite number of steps to approximate an continuous surface. Several authors have explored the theoretical limitations of estimations of slope and aspect calculations on gridded surfaces (e.g., Heuvelink et al., 1989; Hunter and Goodchild, 1997; Florinsky, 1998; Abarbanel et al., 2000; Zhou and Liu, 2004); in general, TE can be related directly to the grid-sampling – more tightly sampled grids will deviate less from the original surface.





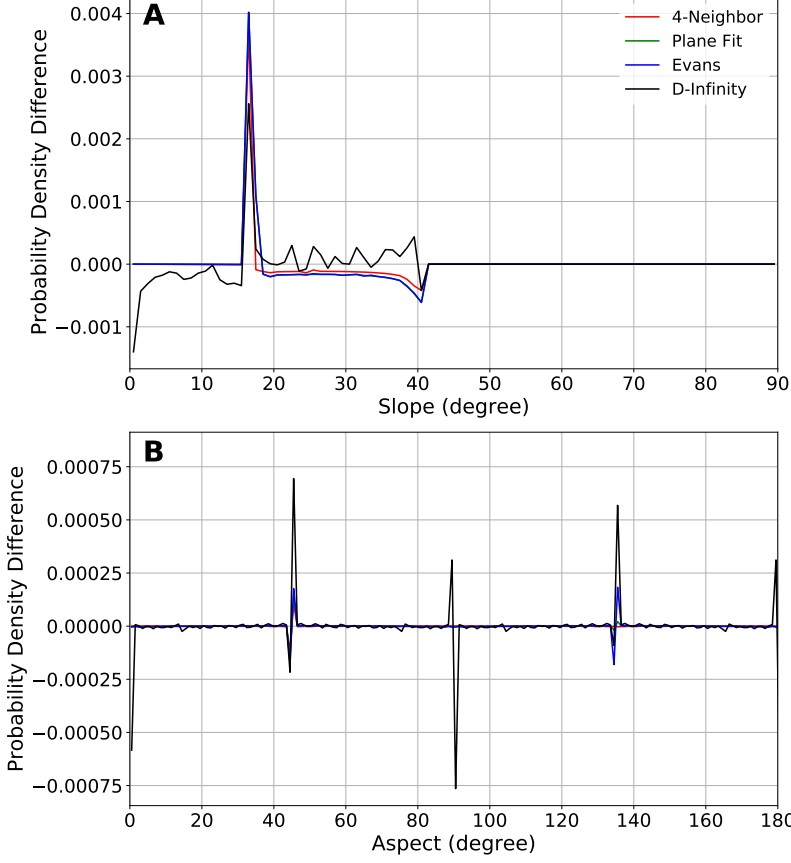

**Figure 1.** (A) Slope and (B) aspect distribution differences from analytical solutions of slope and aspect on a Gaussian Hill, on one-degree bins. None of the methods perfectly match the analytical solution, with slightly different offset magnitudes. In particular, each method produces aspect errors at the cardinal directions.

In addition to TE, real-world DEMs will have some degree of measurement uncertainty. The magnitude of that uncertainty varies across data collection methods and data post-processing, but also depends on the terrain itself and is often difficult to estimate (Heuvelink et al., 1989; Lee et al., 1992; Bolstad and Stowe, 1994; Florinsky, 1998; Fisher, 1999; Smith and Sandwell, 2003; Farr et al., 2007; Wechsler and Kroll, 2006; Wechsler, 2007; Mukul et al., 2017; Purinton and Bookhagen, 2017, 2018; Wessel et al., 2018). Any uncertainty in elevation estimates will propagate into the calculated topographic metrics.

5    The impacts of TE and DEM uncertainty on slope and aspect estimations can be quantified for any gridded data. We first use synthetic data with known properties as a control to define generalized functions applicable to any DEM, and to develop a quality metric for slope and aspect calculations. Following this analysis, we turn to a high-resolution lidar dataset covering complex terrain to study the spatial structure of uncertainty in topographic metrics propagating from both TE and DEM uncertainty. This novel approach allows us to identify the optimal grid resolution that minimizes the error bounds from the



10  combination of TE and PEU, and to analyze the implications of using sub-optimal DEM resolutions for calculating slope and

aspect.

## 2   Data and Methods

In this analysis, we demonstrate the limitations of calculating slope and aspect using synthetic data surfaces. These surfaces

serve as a valuable control dataset, as the precise analytical values for slope and aspect at each grid cell are known. We further

add normally distributed random noise with a known mean and standard deviation to our synthetic data to investigate the

impacts of DEM uncertainty on topographic metrics. Following the discussion of synthetic data, we apply the same methods

to high-resolution lidar data interpolated to DEMs with varying spatial resolutions.

### 2.1   Synthetic Data

We generate regular $n \times n$ grids of size $n$=11, 101, 1001 describing a Gaussian hill with a maximum height of one, with

elevations defined as $z = e^{(-x^2 - y^2)}$ (Figure 2). A second surface describing a sphere is included in the Supplement. It is

important to note that the horizontal and vertical units for the synthetic data are arbitrary – the grid cell spacings generated by

using differently sized grids are an expression of the width-to-height ratio of the surface. The python functions used to generate

our synthetic surfaces can be found on GitHub: https://github.com/UP-RS-ESP/TopoMetricUncertainty.

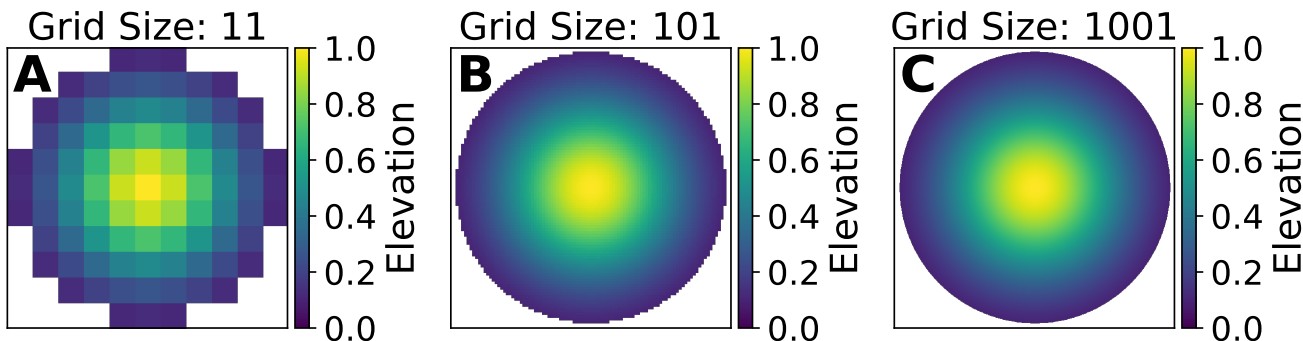

**Figure 2.** Gaussian hill elevations at $n$=11, 101, and 1001. As grid sizes increase, a smoother surface is generated. Note that the horizontal

and vertical units are arbitrary and that the grid-cell spacings are an expression of the width-to-height ratio of the surface. Cross sections of

the shapes at all grid sizes can be seen in Supplemental Figure 1.

### 2.2   Topographic Metrics

In this study, we focus on the topographic metrics slope $S(x, y)$ and aspect $A(x, y)$ which we define as,

$$S(x,y) = \frac{180°}{\pi} \times \arctan \sqrt{\left(\frac{\partial z}{\partial x}\right)^2 + \left(\frac{\partial z}{\partial y}\right)^2} \tag{1}$$



and

$$A(x,y) = \frac{180°}{\pi} \times \arctan\left(\frac{\partial z}{\partial y} \times \frac{\partial x}{\partial z}\right) + 180° \qquad (2)$$

with $x$ and $y$ being the spatial coordinates. Key in calculating slope and aspect is the calculation of the directional derivatives $\frac{\partial z}{\partial x}$ and $\frac{\partial z}{\partial y}$.

There exist a wide range of methods for calculating the directional derivatives on gridded data that broadly fall into three
classes: (1) four-neighborhood methods, (2) eight-neighborhood methods, and (3) steepest descent methods (e.g. Evans, 1980; Zevenbergen and Thorne, 1987; Skidmore, 1989; Tarboton, 1997; Dunn and Hickey, 1998; Schmidt et al., 2003; Zhou and Liu, 2004; Oksanen and Sarjakoski, 2005). In our analysis, we use the standard second-order finite difference approximation as implemented in Python Numpy (Fornberg, 1988; Durran, 1999), which is a four-neighborhood method. We tested additional methods using eight-neighborhoods (e.g., singular-value decomposition plane fitting and the method of Evans (1980)) – which
provide some underlying DEM-smoothing and are well-suited to noisy DEM data – and using steepest descent (e.g., D-8 and D-Infinity (Tarboton, 1997)), which identify the steepest slope that water would take and are optimized for hydrologic use cases. However, these methods do not necessarily provide the most representative slope of the terrain, and show distinct differences from the analytical solutions of slope and aspect (cf. Supplemental Figures 2 and 3). We choose to use the simple four-neighbor method in our analysis as it results in the lowest relative error when compared with the analytical solution for
slope and aspect (Supplemental Figure 2) and does not include any smoothing of the underlying DEM.

## 3 Sources of Uncertainty in Topographic Metrics

Biases in topographic metrics can be determined both numerically, by comparing the results of calculations to known values, and analytically, by deriving the impacts of both TE and DEM uncertainty on topographic metrics.

### 3.1 Truncation Error

We derive the TE from the formulas of topographic metrics by propagating the uncertainty of the second-order finite difference approximation for the directional derivatives $\frac{\partial z}{\partial x}$ and $\frac{\partial z}{\partial y}$. Neglecting rounding errors, we assume the uncertainty $\varepsilon$ of the second-order finite difference to be bounded by

$$\left|\frac{\partial z}{\partial a} - \frac{z(a+h) - z(a-h)}{2h}\right| \lesssim \frac{h^2}{6}\left|\frac{\partial^3 z}{\partial a^3}\right| = \varepsilon_a \qquad (3)$$

where $a$ represents either $x$ or $y$, and $h$ is the grid spacing.



The considered topographic metrics slope $S(x,y)$ and aspect $A(x,y)$ depend on both directional derivatives $\frac{\partial z}{\partial x}$ and $\frac{\partial z}{\partial y}$. As the corresponding uncertainties $\varepsilon_x$ and $\varepsilon_y$ are not independent, propagated TE $T$ for slope ($T_S$) and aspect ($T_A$) are given by

$$|T_S(x,y)| = \sqrt{\left(\partial_{\frac{\partial z}{\partial x}}S\right)^2 \varepsilon_x^2 + \left(\partial_{\frac{\partial z}{\partial y}}S\right)^2 \varepsilon_y^2 + 2\left(\partial_{\frac{\partial z}{\partial x}}S\right)\left(\partial_{\frac{\partial z}{\partial y}}S\right)\varepsilon_x\varepsilon_y} \tag{4}$$

$$|T_A(x,y)| = \sqrt{\left(\partial_{\frac{\partial z}{\partial x}}A\right)^2 \varepsilon_x^2 + \left(\partial_{\frac{\partial z}{\partial y}}A\right)^2 \varepsilon_y^2 + 2\left(\partial_{\frac{\partial z}{\partial x}}A\right)\left(\partial_{\frac{\partial z}{\partial y}}A\right)\varepsilon_x\varepsilon_y} \tag{5}$$

These can be more explicitly expressed using Equation (3) as

$$5 \quad |T_S(x,y)| = \frac{180°}{\pi} \times \sqrt{\frac{(\frac{\partial z}{\partial x})^2\varepsilon_x^2 + (\frac{\partial z}{\partial y})^2\varepsilon_y^2 + 2\frac{\partial z}{\partial x}\frac{\partial z}{\partial y}\varepsilon_x\varepsilon_y}{\left(\frac{\partial z}{\partial x} + \frac{\partial z}{\partial y}\right)\left(1 + \frac{\partial z}{\partial x} + \frac{\partial z}{\partial y}\right)^2}} \tag{6}$$

$$|T_A(x,y)| = \frac{180°}{\pi} \times \sqrt{\frac{(\frac{\partial z}{\partial x})^2\varepsilon_x^2 + (\frac{\partial z}{\partial y})^2\varepsilon_y^2 - 2\frac{\partial z}{\partial x}\frac{\partial z}{\partial y}\varepsilon_x\varepsilon_y}{\left(\frac{\partial z}{\partial x}\right)^4\left(1 + \frac{\partial z}{\partial x} + \frac{\partial z}{\partial y}\right)^2}} \tag{7}$$

The sign of the TE is derived from symmetry considerations in the following way:

$$\text{sign}(T_S) = \text{sign}\left(\arctan\sqrt{\left(\frac{\partial z}{\partial x} + \varepsilon_x\right)^2 + \left(\frac{\partial z}{\partial y} + \epsilon_y\right)^2}\right.$$
$$\left. - \arctan\sqrt{\left(\frac{\partial z}{\partial x} - \varepsilon_x\right)^2 + \left(\frac{\partial z}{\partial y} - \epsilon_y\right)^2}\right) \tag{8}$$

and

$$10 \quad \text{sign}(T_A) = \text{sign}\left(\arctan\frac{\frac{\partial z}{\partial y} + \varepsilon_y}{\frac{\partial z}{\partial x} + \epsilon_x} - \arctan\frac{\frac{\partial z}{\partial y} - \varepsilon_y}{\frac{\partial z}{\partial x} - \epsilon_x}\right) \tag{9}$$

Equations 6-9 allow us to calculate the magnitude of the TE at each point $(x,y)$ on our synthetic surfaces (Figure 3).



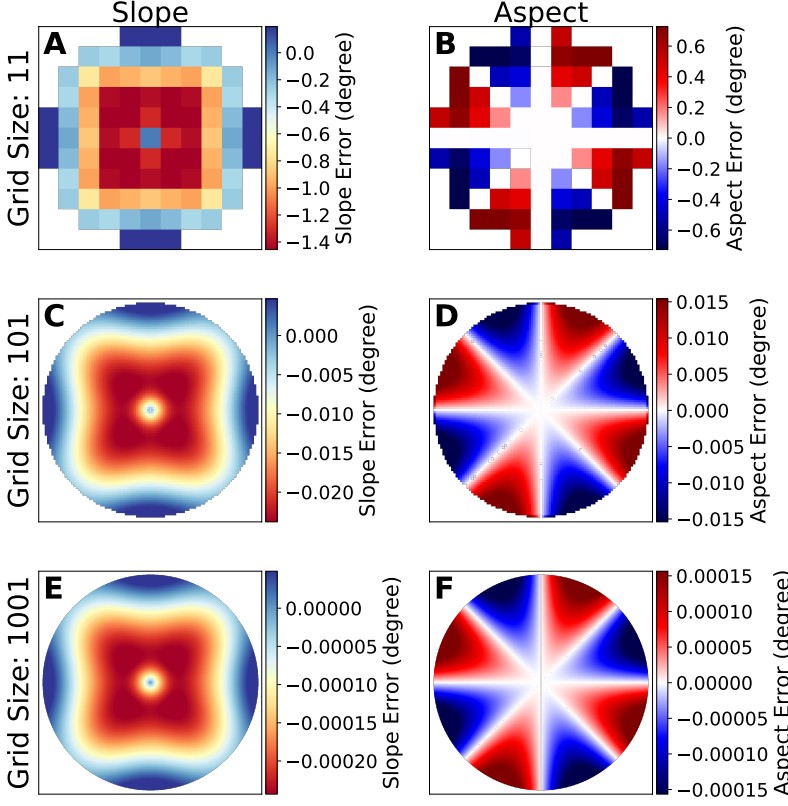

**Figure 3.** Gaussian hill slope (A, C, E) and aspect (B, D, F) errors from the analytical TE. All grid sizes show clear spatial patterns in slope and aspect errors. Offset magnitudes scale with grid size. Errors on the $n$=1001 grid (bottom row: E, F) are generally two orders of magnitude smaller than those for the $n$=101 grid (middle row: C, D). Colors scaled from 5th to 95th percentiles of error magnitudes.

As can be seen in Figure 3, the magnitude of the slope and aspect error decreases with increasing $n$. This implies that as more tightly sampled models of a surface are used, the accuracy of slope and aspect estimations increases. This follows the work of Abarbanel et al. (2000), which shows that finite-difference derivative estimations have error magnitudes that scale with the inverse of the grid spacing for evenly spaced grids. However, even with increasingly accurate surface models, there remains some TE simply due to the fact that a smooth surface cannot be perfectly represented by gridded data. At very fine scales, computer rounding will also lead to errors in slope and aspect calculations. In this study, however, we do not use fine enough grids for this to impact our error estimates.

As the precise mathematical definitions of our synthetic surfaces are known, we can analytically derive their slope and aspect values at each point and compare those results to numerical calculations (see Supplemental Figure 4). In the absence of DEM uncertainty, the difference between analytically- and numerically-derived topographic metrics will be dominated by the TE from the second-order finite difference approximation. Additionally, modifying the absolute magnitude of the shape





heights (e.g., a maximum height of 100 instead of one), does not modify the absolute magnitudes of slope and aspect TE (see Supplemental Figure 5).

While the magnitudes of the errors are small, they have important impacts on distributions drawn from larger grids (e.g., entire surfaces, see Figure 4). In the case of aspect, the alternating spatial pattern of its TE pushes aspect values towards the cardinal directions – leading to distinct positive and negative spikes in the aspect distribution. This error is not confined only

to the cardinal directions – there exists symmetry across the aspect distribution that is easiest to see with very fine binning (see Supplemental Figure 6).

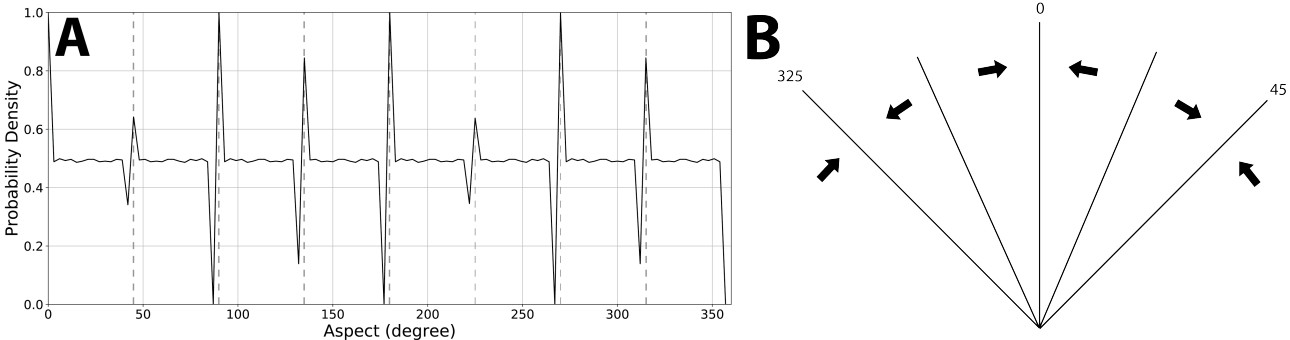

**Figure 4.** (A) Density histogram of synthetic data ($n$=1001) using the four-neighbor method, on one-degree bins. Spikes clearly occur around the cardinal directions. The symmetry across the aspect distributions can be further explored in the Supplemental Figure 5 with very fine binning. (B) Conceptual diagram of aspect errors. Aspect values are pushed towards and away from certain angles (0/45/90/etc).

Slope and aspect calculations have very differently spatial error patterns (see Figure 3). However, as with aspect errors, slope errors depend on the orientation of the grid despite the radial symmetry of the surface. This is easiest to see in the slope difference map of the Gaussian hill, where error magnitudes are not simply scaled by slope (see Figure 3E).

## 3.2   DEM Uncertainty

Elevation models are never perfect – there are always errors and uncertainties due to data collection or processing. The uncertainty in calculated topographic metrics can be constrained by propagating DEM uncertainties into their calculations (e.g. Heuvelink et al., 1989; Hunter and Goodchild, 1997; Zhou and Liu, 2004; Oksanen and Sarjakoski, 2005); we refer to the uncertainty introduced into slope and aspect calculations from elevation uncertainty as propagated elevation uncertainty (PEU).

For our synthetic data, we assume the simplistic but straight forward error model of spatially uncorrelated white noise. We define noise to be normally distributed and consider two cases: (1) homogeneous white noise, i.e., noise drawn from one unique normal distribution $\mathcal{N}(\mu = 0, \sigma^2 = const)$, and (2) slope dependent, spatially varying white noise, i.e., noise drawn from a family of normal distributions $\mathcal{N}(\mu = 0, \sigma_S^2)$. Since for each grid cell noise is drawn independently, the PEU for slope ($E_S$) and aspect ($E_A$) is given by





$$E_S = \sqrt{\sum_{i \in \{x-h, x+h\}} \sum_{j \in \{y-h, y+h\}} \left(\frac{\partial S}{\partial z_{ij}}\right)^2 \sigma_{z_{ij}}^2} \tag{10}$$

$$E_A = \sqrt{\sum_{i \in \{x-h, x+h\}} \sum_{j \in \{y-h, y+h\}} \left(\frac{\partial A}{\partial z_{ij}}\right)^2 \sigma_{z_{ij}}^2} \tag{11}$$

As before, this can be expressed more explicitly using Equation (3) as

$$E_S = \frac{90°}{h\pi} \sqrt{\frac{\left(\frac{\partial z}{\partial x}\right)^2 \left(\sigma_{z_{x+h,y}}^2 + \sigma_{z_{x-h,y}}^2\right) + \left(\frac{\partial z}{\partial y}\right)^2 \left(\sigma_{z_{x,y+h}}^2 + \sigma_{z_{x,y-h}}^2\right)}{\left(\left(\frac{\partial z}{\partial x}\right)^2 + \left(\frac{\partial z}{\partial y}\right)^2\right)\left(1 + \left(\frac{\partial z}{\partial x}\right)^2 + \left(\frac{\partial z}{\partial y}\right)^2\right)^2}} \tag{12}$$

$$E_A = \frac{90°}{h\pi} \sqrt{\frac{\left(\frac{\partial z \partial x}{\partial y \partial z}\right)^2 \left(\sigma_{z_{x+h,y}}^2 + \sigma_{z_{x-h,y}}^2\right) + \sigma_{z_{x,y+h}}^2 + \sigma_{z_{x,y-h}}^2}{\left(\frac{\partial z}{\partial x}\right)^2 \left(1 + \left(\frac{\partial z \partial x}{\partial y \partial z}\right)^2\right)^2}} \tag{13}$$

5    As with TE, PEU introduces distinctive spatial patterns into the slope and aspect estimates (Figure 5). Both slope and aspect
uncertainties are higher on shallow slopes, as has been shown in previous work (Florinsky, 1998; Zhou and Liu, 2004). On
flat or nearly flat terrain, the elevation uncertainty will more strongly impact the estimated partial derivatives, and thus lead to
erroneous slope and aspect estimates. It is also important to note that aspect varies between [0, 360) and slope only between
[0, 90), which is why we expect standard deviations to be different by a factor of four (see Figure 5).

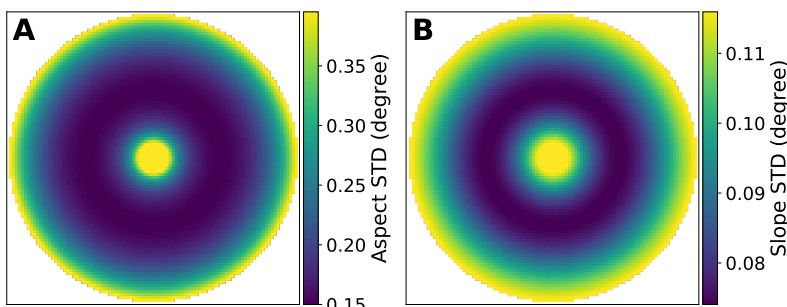

**Figure 5.** Gaussian hill ($n=101$) slope and aspect standard deviations for spatially invariant noise (std=$1e^{-4}$), derived from elevation uncertainty propagation (Equations 12-13). Aspect has higher standard deviations than slope, as aspect varies between [0, 360) and slope only between [0, 90). This is evident especially on flat terrain (e.g., the edges of the Gaussian hill).

10    When we compare our analytically-derived topographic metric uncertainty patterns (Figure 5) to those generated from actual
random noise – using an ensemble of 10,000 synthetic surfaces, each generated with normally distributed noise – the spatial
patterns and magnitudes are nearly identical between the two methods (cf. Supplemental Figure 7).





## 4  Optimal Grid Spacing

Both TE and PEU yield distinct spatial patterns, implying that the error and uncertainty in topographic metrics will vary throughout a DEM. Using both the TE and PEU magnitudes, we can examine whether TE or PEU is dominant at an arbitrary grid spacing, given a known DEM uncertainty (Figure 6).

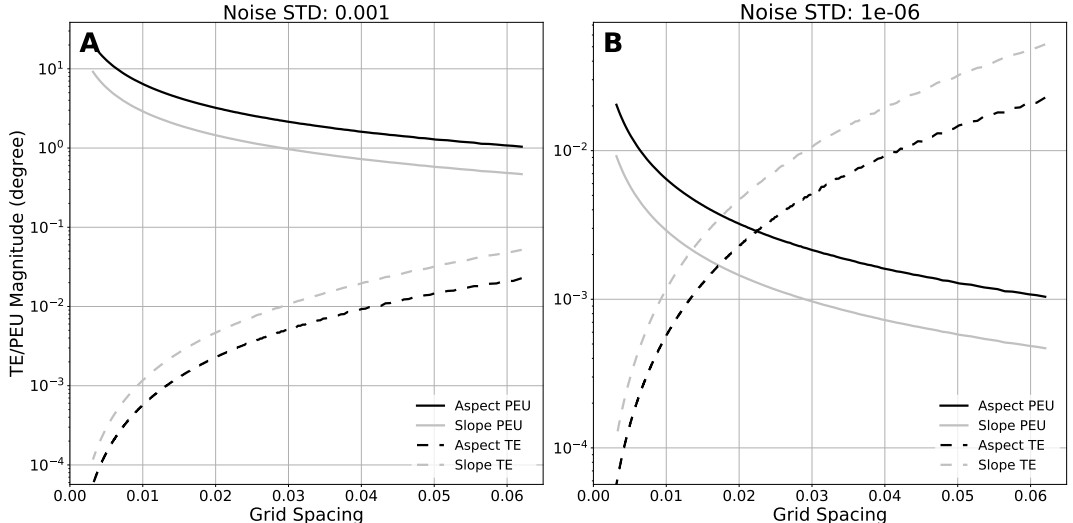

**Figure 6.** TE and PEU magnitudes for two noise levels, (A) $1e^{-4}$ and (B) $1e^{-6}$, compared to grid spacing. Aspect plotted in black, slope plotted in gray. Solid lines for PEU, dashed lines for TE. TE decreases with decreasing grid spacing, while PEU increases as the noise standard deviation relative to grid spacing increases.

As can be seen in Figure 6, TE will approach zero for sufficiently small grid spacings, while PEU will rise as the amount of
5  noise relative to the grid spacing increases. Using both TE and PEU magnitudes across grid spacings, an optimal grid spacing for a given noise level which minimizes their combined influence can be chosen. Note that we do not assume that the quality of the data is maximized when the combination of TE and PEU is minimized, but that the error bounds on topographic metric calculations are minimized. Thus, the optimal grid spacing is preferred for calculating slopes and aspects for use in applications where uncertainty across the entire grid should be minimized.

10  ### 4.1  Metric Quality Ratios

The trade off between TE and PEU can also be thought of as a quality ratio. Simply put, the impact of DEM uncertainty on topographic metrics is modulated by the grid spacing – i.e., 1 cm of vertical noise on a 10 m grid will have a very different impact than 1 cm of vertical noise on a 1 cm grid. As both TE and PEU have distinct spatial patterns, a combined Quality Ratio





(QR) can capture a holistic view of the overlapping influence of TE and PEU. The QR can be defined from the combination of the TE and PEU as:

$$QR = \left( \frac{1}{1 + (m \times |TE|)} \right) \times \left( \frac{1}{1 + (m \times |PEU|)} \right) \tag{14}$$

where $m$ is a normalization factor which accounts for the four-fold difference in the range of possible values between slope and aspect. On a surface without noise, the QR will approach one as grid spacing and TE approach zero. The influence of TE

5   and PEU on calculated QRs can be seen by comparing the same synthetic grid with two noise levels (see Supplemental Figure 8). High noise levels reveal similar spatial patterns to those seen in Figure 5, while low noise levels show the influence of TE (cf. Figure 3).

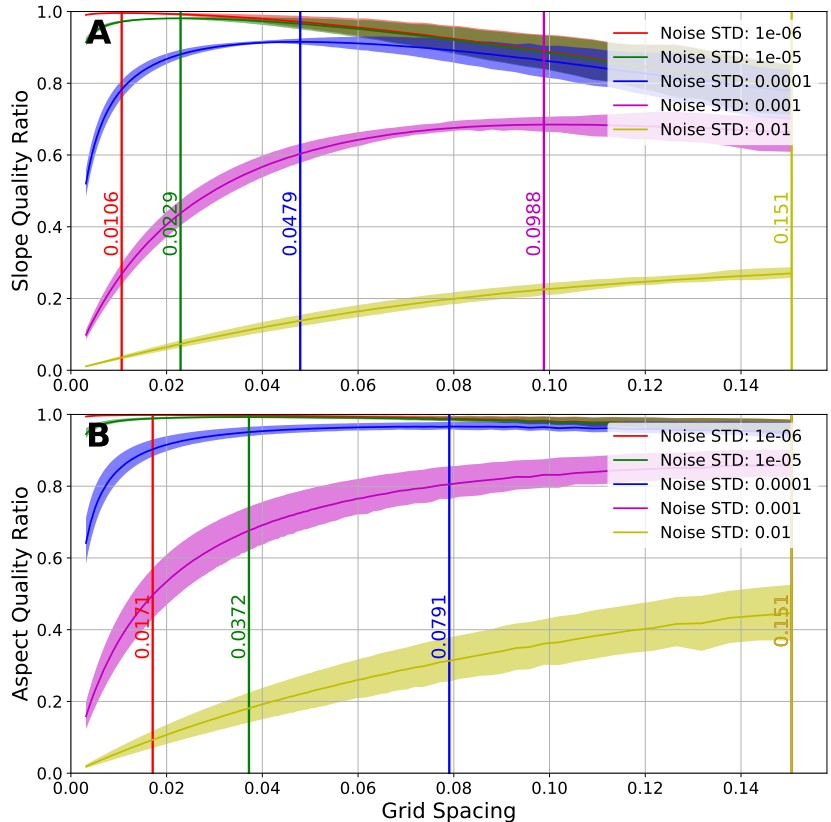

**Figure 7.** Gaussian hill slope (A, top) and aspect (B, bottom) QRs vs grid spacings, for a range of noise levels ($1e^{-2}$ to $1e^{-6}$). 25th-75th percentile QRs shaded for each noise level. Optimal grid spacings (maximum QR) for each noise level are marked with vertical lines. Slope calculations result in lower optimal grid spacings compared to aspect. Higher noise levels lead to higher optimal grid spacings for both slope and aspect calculations. Note that the purple and yellow lines in panel B have the same optimal grid spacing, and thus only the yellow line is visible.



The QR can also used as a normalized metric to compare DEMs across grid spacings and noise levels (Figure 7, and Supplemental Figure 9). Each noise level/grid spacing pair has a point at which the QR of the DEM is maximized. This point represents the optimal trade off between TE and PEU over the entire grid.

As can be seen in Figure 7, as noise levels increase, the optimal grid spacing increases, and the QR of that optimal spacing decreases. For very noisy datasets (purple and yellow lines, Figure 7), very large grid cell spacings – i.e., aggregation – can help reduce the combined influence of TE and PEU on slope and aspect estimates.

There are distinct differences between the optimal slope and aspect grid spacings for the same dataset. For any given noise level, the optimal grid spacing to calculate aspect is higher than that for slope. The difference in optimal grid spacing is driven by the relative magnitudes of the PEU between slope and aspect calculations – the PEU for aspect calculations is always higher than that for slope due to differences in the formulas for slope and aspect (see Figure 6 and Equations 12-13).

## 4.2 Heterogeneous Noise

In practice, datasets will not have homogeneous noise across grid spacings – coarser parameterizations of a surface will include more uncertainty as fine-scale features are aggregated into single pixels. Additionally, noise in real-world datasets is influenced by non-uniform landscape features such as slope, aspect, terrain relief, and vegetation cover (e.g., Kraus and Pfeifer, 1998; Kyriakidis et al., 1999; Holmes et al., 2000; Smith and Sandwell, 2003; Carlisle, 2005; Oksanen and Sarjakoski, 2006; Rodriguez et al., 2006; Shortridge and Messina, 2011; Purinton and Bookhagen, 2017, 2018). While several authors (e.g., Hunter and Goodchild, 1997; Zhang and Goodchild, 2002; Fisher and Tate, 2006) have examined complex error models in real data, we choose to use a simplistic error model based on terrain slope – e.g., higher slopes have higher noise levels – to look at the first-order impacts of heterogeneous noise on the calculation of topographic metrics.

When we compare the heterogeneous noise result to the analysis shown in Figure 7 (cf. Supplemental Figure 10), the results are very similar, albeit with higher optimal grid spacings. We can also compare the optimal grid spacings for homogeneous and heterogeneous noise directly for a set of noise levels from $1e^{-8}$ to $1e^{-2}$ (Figure 8).

As can be seen from Figure 8, slope can be calculated on a smaller grid spacing with slope-biased noise, while aspect should be calculated with a larger grid spacing. The divergence in the optimal grid spacings between homogeneous and heterogeneous noise are driven by the differences in the influence of terrain slope on slope and aspect QRs. While both slope and aspect calculations have similar grid-spacing dependent TE magnitudes, the differences in the relative magnitudes of the PEU lead to different responses to the distribution of noise on the DEM (see Figure 6). The simplistic slope-dependent error model used here is unlikely to translate to a real-world setting; however, it is clear that the spatial distribution of noise on the DEM influences the choice of optimal grid spacing for both slope and aspect calculations.





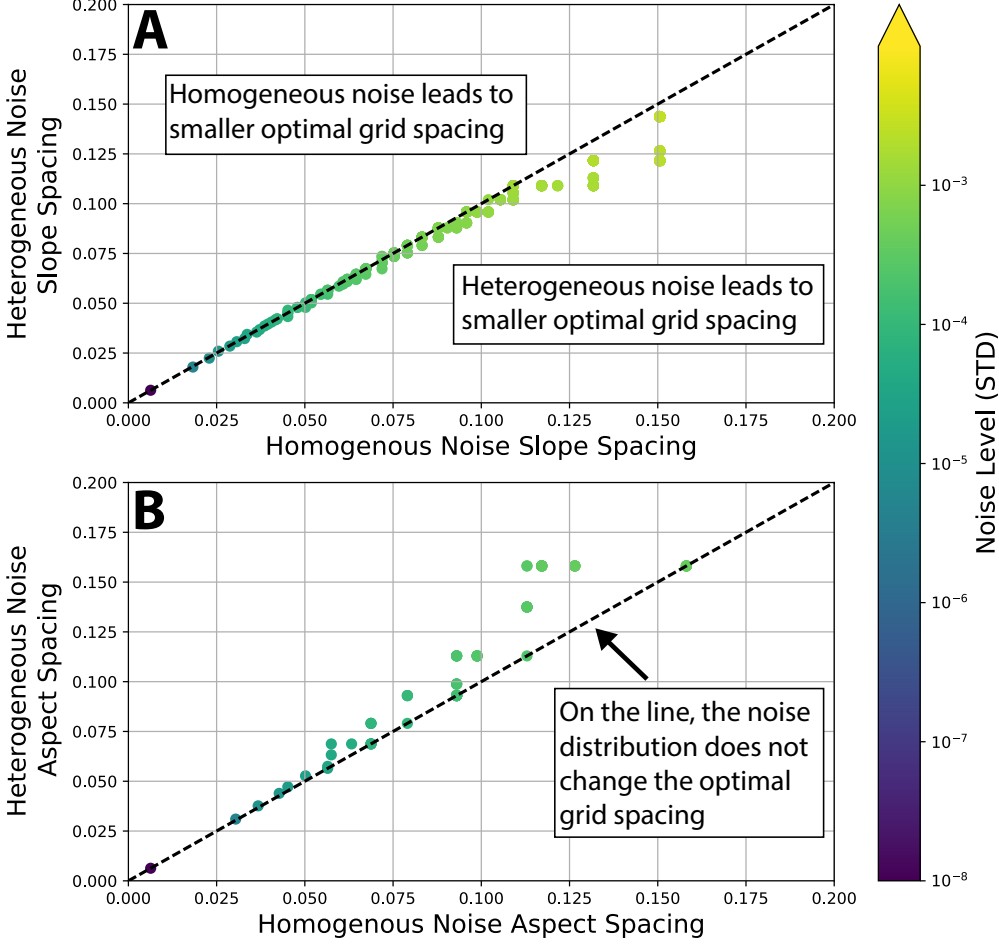

**Figure 8.** Optimal grid spacing for slope (A) and aspect (B) across a range of noise levels ($1e^{-2}$ to $1e^{-8}$). Points in the upper triangle have smaller optimal grid spacings with homogeneous noise. Points in the lower triangle have smaller optimal grid spacings with heterogeneous noise. Points on the dashed line have the same optimal grid spacings regardless of the noise distribution. Heterogeneous noise leads to smaller slope grid spacings but larger aspect spacings for the same average noise level.

## 5   Impacts of Noise on Topographic Distributions

As can be seen in Figures 5-7, the introduction of noise has non-trivial impacts on slope and aspect calculations. As was evident in Figures 1 and 4, TE can significantly modify the ensemble distributions of topographic metrics. To examine the impact of both TE and PEU on the frequency distributions of slope and aspect, we generate 1,000 member ensembles of slope and aspect for each combination of two different grid sizes ($n$=101, 1001) and three different noise levels (homogeneous noise, std=$1e^{-2}$, $1e^{-3}$, $1e^{-4}$). We then bin the ensemble data into one-degree slope and aspect bins.





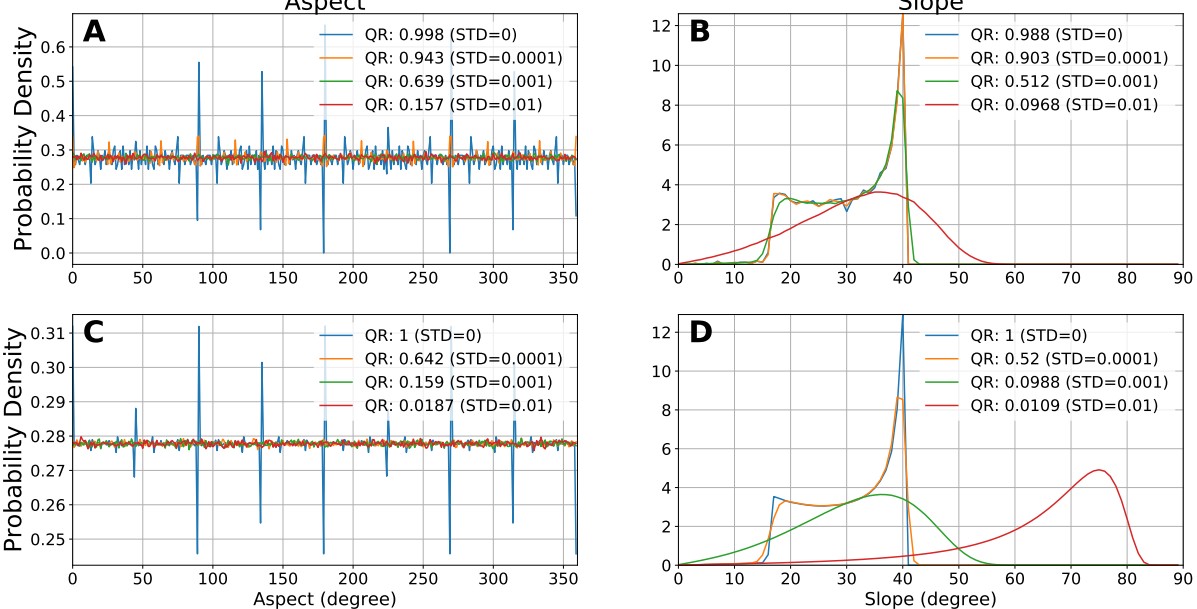

**Figure 9.** Aspect (A, C) and Slope (B, D) probability distributions (one-degree bins) for a Gaussian hill with homogeneous noise at variable levels (std=0, $1e^{-2}$, $1e^{-3}$, $1e^{-4}$) for $n$=101 (A, B) and $n$=1001 (C, D). The introduction of small levels of noise washes the spikes out of the aspect distribution. Slope distributions are steeper with the introduction of higher noise levels. Similar density functions are generated at similar QRs, despite differences in absolute noise levels.

When homogeneous noise is added to synthetic surfaces, the large spikes in aspect distributions related to points moving in and out of cardinal directions (e.g., at 45/90/135 degrees, see Figure 4) are smoothed out into neighboring bins. However, this does not imply that the calculated aspects are 'better', just that they have a more even distribution – the PEU overprints the TE as seen in Figure 3. This can clearly be seen in Supplemental Figure 11, which shows the impact of noise on slope and aspect grids.

5    Slope distributions are systematically shifted towards higher values as more noise is added to the synthetic surfaces. This effect is due to the presence of more large 'steps' in the synthetic data, where smooth transitions across elevation gradients are replaced with more stepped hills and dips. Across noise levels and grid sizes, distributions with similar QRs maintain the same general shape. This indicates that the QR captures how 'wrong' the aggregate distribution is when compared to the original synthetic surface.

10    If different window sizes are used to calculate slope and aspect (e.g., 5x5 and 7x7), slope estimates are higher and the aspect 'spikes' at cardinal directions are slightly diminished. Larger window sizes somewhat – but not completely – remove the impacts of noise on aspect distributions. They do not, however, help minimize differences in the slope distributions (see Supplement Figure 12).



# 6 Case Study: Multi-Resolution Lidar on Santa Cruz Island

## 6.1 Dataset Description

The Santa Cruz Island (SCI, see Figure 10) is part of the Channel Islands off the coast of south-central California. It is a tectonically active island (Neely et al., 2017) with steep topography and pronounced erosion (Perroy et al., 2010, 2012), diverse lithologic cover (Dibblee, 2001), and moderate and species-rich vegetation cover (Baguskas et al., 2014). The lidar point cloud data used to derive DEMs at a range of spatial resolutions were obtained from the 2010 US Geological Survey Channel Islands Lidar Collection (OpenTopography, 2012). The point cloud has an average point ground-classified density of ~10 points/m².

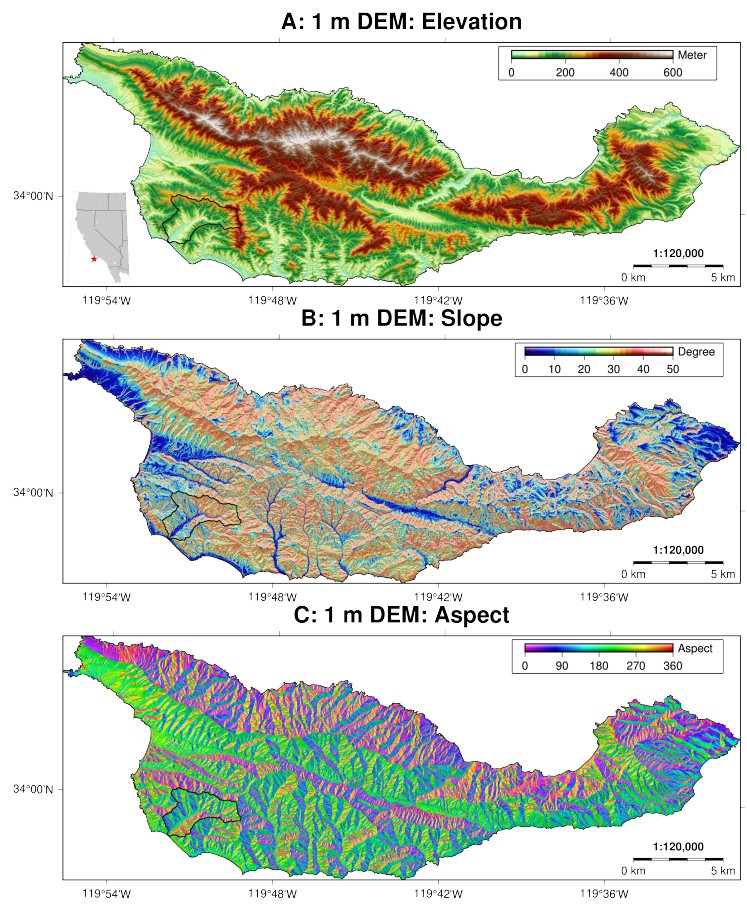

**Figure 10.** Topographic setting of Santa Cruz Island (SCI), showing elevation (A), slope (B) and aspect (C) derived from a 1-m DEM and calculated from a total of $2.3 \times 10^9$ ground-classified lidar points. The SCI covers a wide range of terrain types, slope regimes, and all aspect directions. Black polygon (southwest SCI) indicates the Pozo catchment shown in Figure 11. For a canopy height model, we refer to Supplemental Figure 13.



## 6.2 Deriving Elevations and Uncertainties from Point Cloud Data

Using LAStools (LAStools, 2017), we use a triangulated irregular network approach to derive average point cloud elevations at each grid cell center, for each of our chosen grid resolutions (1-30 meters, in one meter steps, and 90 meters). We have tested simpler interpolation schemes, including the lowest elevation and the central location for each grid cell, but the resulting uncertainties in topographic metrics remain the same. Thus, we decided to use the more commonly applied triangulated irreg-

ular network approach. In addition to producing DEMs, we generate pixel-wise standard deviations estimated from the point cloud. That is, we determined the standard deviation of all ground-classified lidar points falling into each grid cell for each grid resolution. While vegetation cover may impact ground-classification results, our calculations are based on ground-classified points only. Because of the high point-cloud density, this standard deviation metric should reflect terrain variability and not vegetation cover. This measure serves as a DEM uncertainty and is highly spatially heterogeneous. Elevation and standard

deviation maps for additional grid resolutions can be found in the Supplement (Supplemental Figures 14-16). An island-wide canopy-height model can be found in Supplemental Figure 13.

## 6.3 Mapping Uncertainty on SCI

Using our standard deviation grids, we can then calculate both TE and PEU on SCI (see Supplemental Figure 17). Using these two grids as inputs, we can also derive the QR across the entire SCI (Figure 11). Note that maximal QRs are low when compared

to the theoretical models shown in Figures 7 and 9; this is due to both the much higher inherent noise in the real-world data, and to the more complex terrain represented by the lidar data.



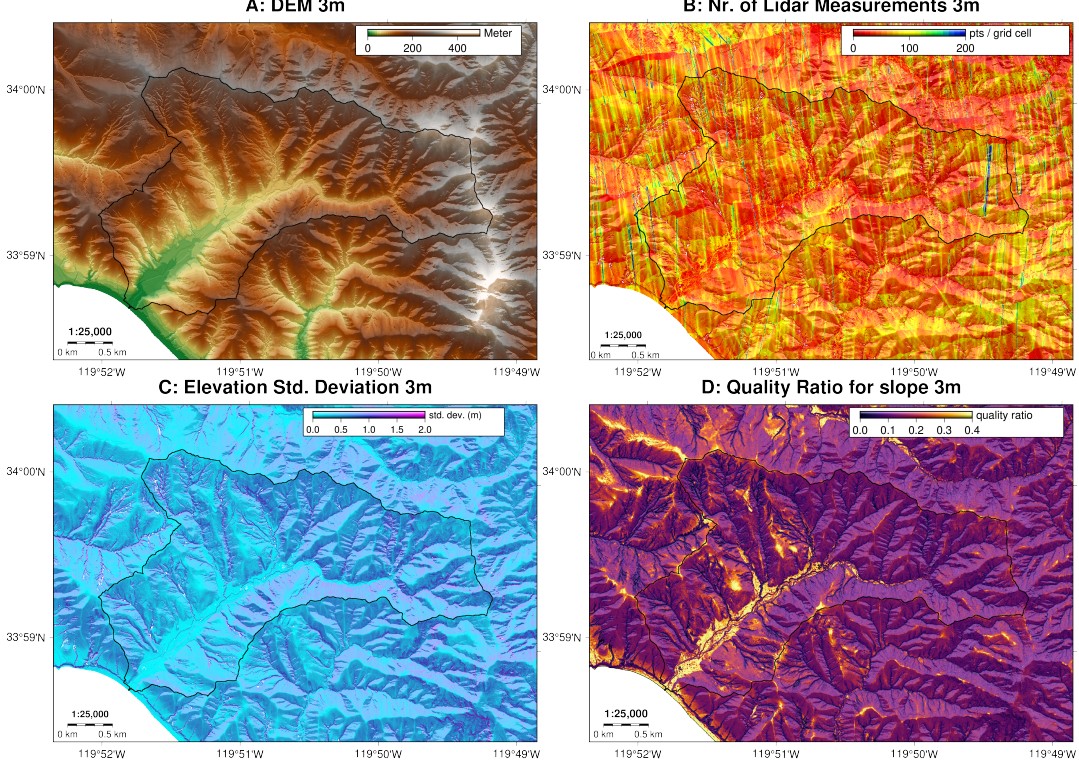

**Figure 11.** Selected metrics for the Pozo catchment in the southwestern part of SCI, calculated on the optimal grid resolution for this catchment (three meter grid). (A) Elevation, (B) lidar point density, (C) elevation standard deviation, (D) slope quality ratio (see Equation 14). The aspect quality ratio is shown in Supplemental Figure 18. A comparison of the elevation standard deviation and the lidar canopy height model is shown in Supplemental Figure 19.

## 6.4 Identification of Optimal Grid Resolution

Using our pixel-wise standard deviations – and derived QRs – at each grid resolution, we can determine the island-wide optimal grid resolution for the calculation of topographic metrics which minimizes the combined influence of TE and PEU on our lidar dataset.

Unlike the synthetic data examined in Figure 7, the optimal grid resolution for aspect calculations is the same as that

5    for slope. This is due to the distribution of both elevations and elevation uncertainties in the SCI DEM, and the different responses of slope and aspect calculations to the diverse terrain on SCI. Whole-island median TE, PEU, and QR can be found in Supplemental Table 1 for grid spacings from one to ten meters. While four meters is the calculated optimal grid resolution, the QR peak is quite flat (see Figure 12), implying that both three-meter and five-meter DEMs will yield slope and aspect results of nearly equal quality. Indeed, the TE and PEU magnitudes are similar for both metrics between three and five meters

10    (see Supplemental Table 1).



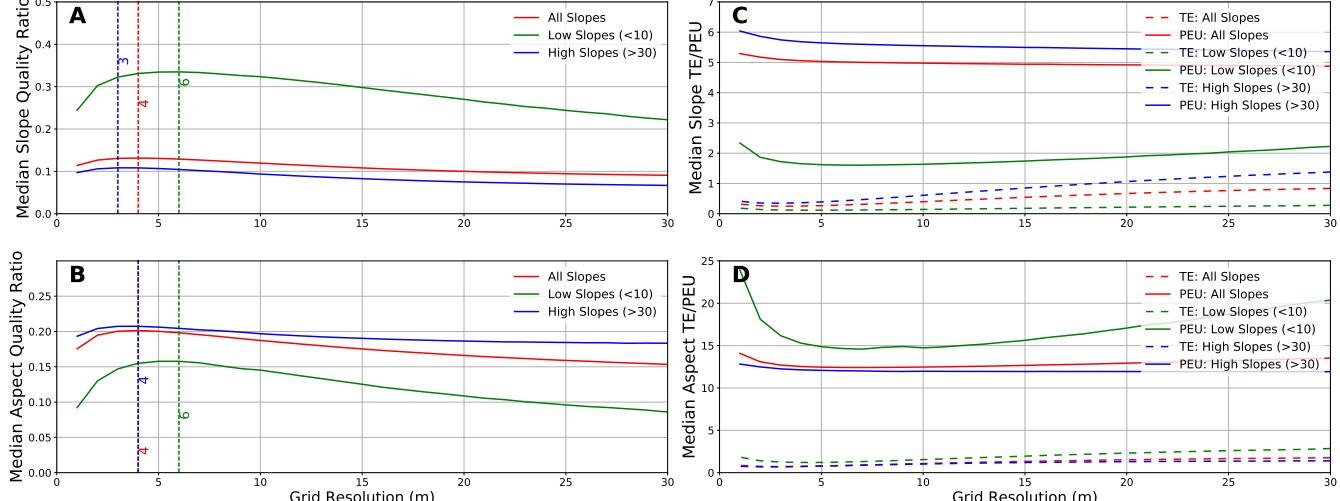

**Figure 12.** Median Quality Ratios (QRs) for all slopes, low slopes (<10) and high slopes (>30) across grid resolutions (1-30m). There is a clear trend where low slopes have better slope QRs, but worse aspect QRs. The optimal grid resolution for SCI is four meters for both slope and aspect. For inter-quartile ranges on the QRs, see Supplemental Figure 20.

It should be noted that a four-meter optimal grid resolution is likely too large for many applications, particularly those interested in micro-topography or other small-scale features. In those cases, a higher-resolution DEM should be used with caution – PEU will continue to grow as grid spacing shrinks, and the rate of that increase will accelerate (see Figure 6). This effect will occur even if the DEM uncertainty decreases at smaller scales simply due to the larger impacts of PEU at very small grid spacings. This means that the uncertainty bounds on slope and aspect estimates on very high resolution DEMs will increase quickly, which may limit some analyses. For high-resolution analyses, very high quality DEMs are required.

In this analysis of the airborne lidar dataset from SCI, we find that a four-meter DEM resolution minimizes the combined influence of TE and PEU. Slope and aspect analyses performed at this grid resolution have the smallest error bounds, and thus provide the most reliable data for analysis of topographic metrics over the entire island.

While a single, whole-island, grid resolution is useful in some applications, it is clear from Figure 11 that there are large spatial variations in the DEM uncertainty and the resulting QR, and hence in the optimal grid resolution. We can extend our method of identifying the optimal grid resolution on a catchment-by-catchment basis (Figure 13, map view in Supplemental Figure 21).

There exist large spatial variations in the optimal grid resolution across SCI, driven by differences in terrain slope and DEM quality. When these differences are compared to median catchment slope and median catchment standard deviation, a clear pattern emerges (Figure 13). While there remains significant scatter in the data, catchments with high median slopes tend to have both higher standard deviations and lower optimal grid resolutions. The data is bimodal, with the majority (83% for slope, 95% for aspect) of catchments having optimal resolutions below ten meters.



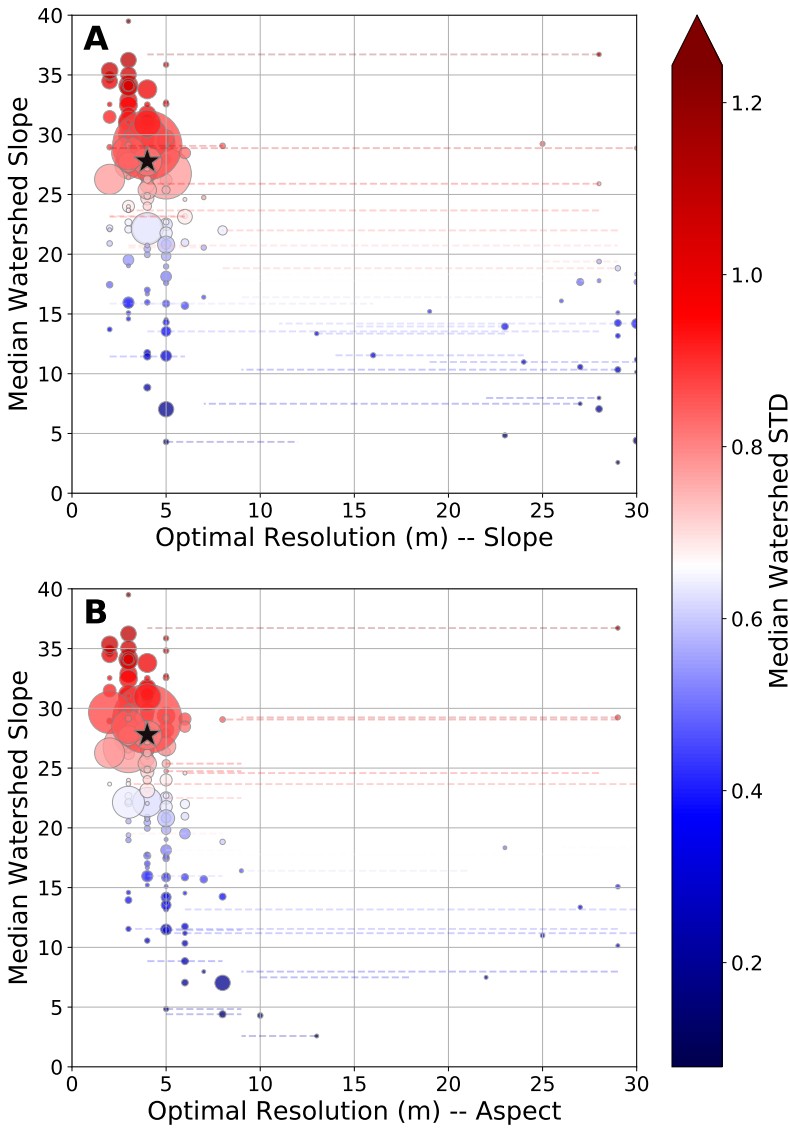

**Figure 13.** Optimal grid resolution for minimizing error bounds on slope and aspect calculations across SCI compared to catchment median slope. Point sizes are scaled to catchment size, from 0.1 to 34 sq km. Dashed lines show the 25th percentile to 75th percentile of optimal grid resolution, colored by the standard deviation of that point. Stars show the whole-island optimal resolution and median slope. There exists a clear trend where lower median slopes lead to lower optimal grid resolutions. Standard deviations also tend to be smaller in these catchments, due to higher DEM fidelity over flat terrain. Catchments with high (>10 meter) optimal grid spacings are small and have complex topography. For a map view of these results, we refer to Supplemental Figure 21.

It is worth noting that the vast majority of the catchments with optimal slopes and aspects above ten meters are small catchments concentrated on the northwestern edge of SCI (see Supplemental Figure 21). These catchments have two unique





topographic features which contribute to the high optimal grid sizes. (1) They have two distinct slope regimes: one steep hill of nearly constant slope, followed by one flatter residual marine terrace with a different constant slope. (2) The catchments drain into steep, deeply incised, cliffs. Interestingly, neither removing the steep cliffs (which have high elevation uncertainty), nor aggregating several small catchments into one larger analysis unit (to improve the statistical reliability of the median QR) result in smaller optimal grid spacings. We posit that the high optimal grid spacings are a product of the unique two-stage topography

in this region. However, an in-depth analysis of the specific factors which drive the optimal terrain resolution in each individual zone is outside of the scope of this manuscript. In applying this method of deriving optimal grid spacings, it is important to take into account the possibility of large differences in the optimal grid spacing over relatively small spatial scales.

### 6.5   Island-Wide Slope and Aspect Distributions

As has been shown in Figures 7 and 12, there is clearly an optimal spatial resolution for the calculation of slope and aspect on

our SCI lidar dataset. Furthermore, in Figure 9, it was shown that changes in QR can have a substantial impact on the slope and aspect distributions of a synthetic surface. This analysis can be extended to the SCI lidar data at a range of spatial scales (Figure 14).

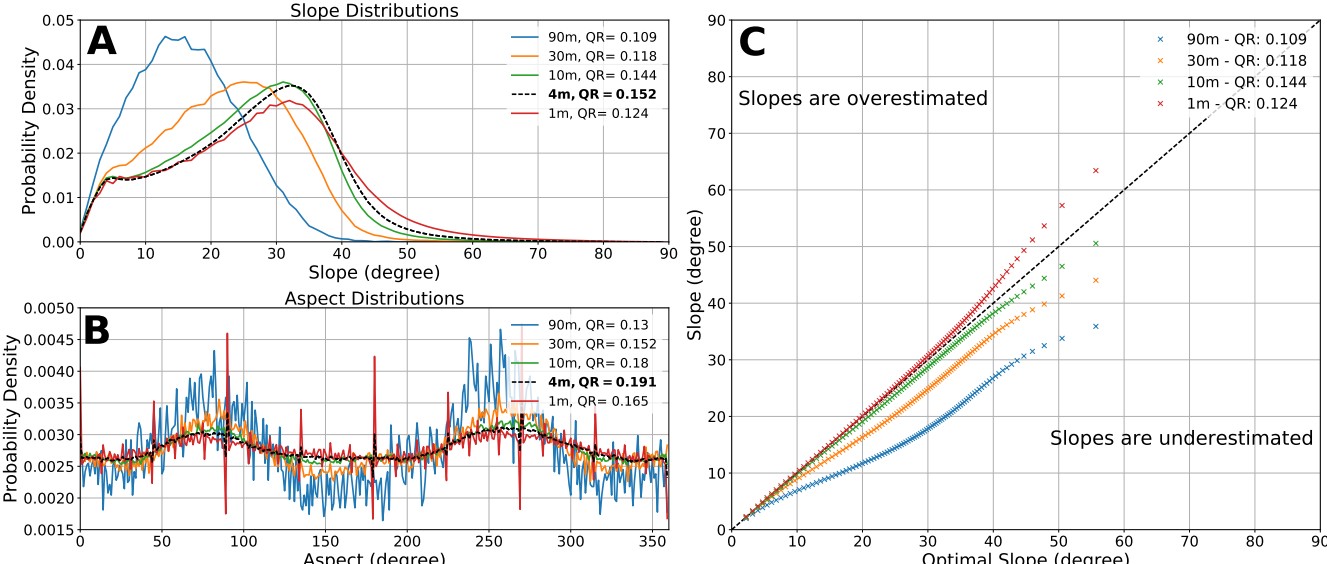

**Figure 14.** (A) Slope and (B) aspect differences across SCI DEMs, with QR calculated using SCI DEM elevation standard deviations (see Figure 11 and Supplemental Figures 13-15). Optimal slope and aspect distributions plotted in black. (C) Quantile-quantile plot showing difference between the entire SCI slope distribution vs the 'optimal' (four meter) resolution. We observe a clear pattern in slope distributions where resolutions below the optimal resolution tend to overestimate slope, those below tend to underestimate slope.

There are clear differences in the slope and aspect distributions with increasingly poor QRs. This effect can also be seen in quantile-quantile plots of the entire SCI slope distribution when compared to the optimal grid resolution (see Figure 14C). Grid

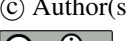



resolutions higher than the optimal result in slope distributions that are dominated by higher slopes, while lower resolutions tend towards lower slopes. This effect can be seen even when the grid resolution is only slightly removed from the optimal resolution (Supplemental Figure 22). Notably, grid resolutions close to the optimal (e.g., three meters and five meters), have QRs and slope distributions that are very similar to the optimal four meter distribution.

As a final test of grid-resolution dependence, we resampled several grids to nominally match the optimal grid resolution – in

5  this case we chose three meters for the Pozo catchment (see Figure 10, Supplemental Figure 23) as the reference distribution. As can be seen in Figure 15A, there are differences between test (one to five meter) resolutions and the reference three meter histogram. When the test datasets are resampled to the three meter reference resolution (using bilinear resampling), the histograms do not converge to the reference three meter distribution (Figure 15B). Slope distributions from resampled data should be used with caution; simply resampling elevation data to the optimal grid resolution before calculating slope will not yield

10  better results – the DEM has to be created from the ungridded point data to fully take advantage of optimal gridding for slope calculations.

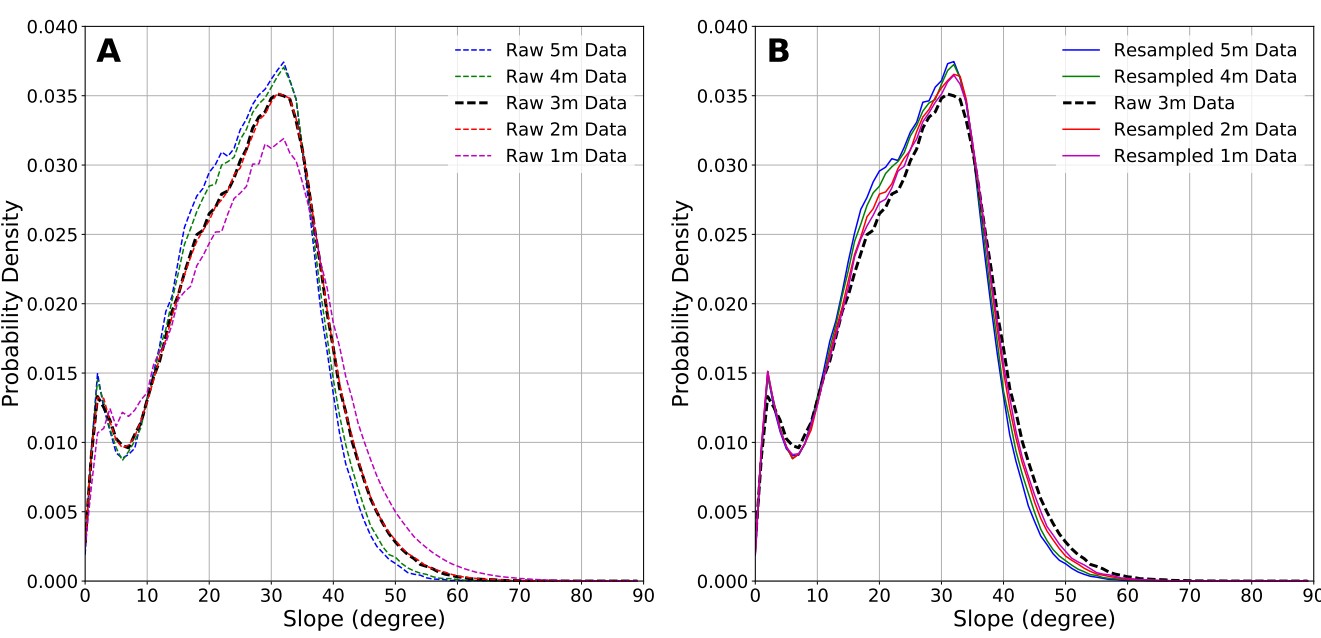

**Figure 15.** (A) Divergence of higher and lower grid-resolution slope distributions from that at the optimal resolution for the Pozo catchment (see Figure 10). Excepting the one meter histogram, the differences between resolutions are small. (B) Slope distributions taken from one, two, four, and five meter data resampled to three meter resolution using bilinear resampling. The resampled data distributions do not converge on the reference three meter distribution.



## 7 Conclusions

This study presents a detailed accounting of uncertainties and errors in the topographic metrics slope and aspect derived from both truncation errors and uncertainty in the underlying source digital elevation model (DEM). We first develop our analysis on synthetic data, which acts as a control dataset. We then extend our methodology onto a high point-density lidar dataset, which allows us to compare the relative impacts of gridding truncation errors and propagated elevation uncertainty on the calculation

of topographic metrics.

From the analysis of both synthetic and real-world data, we identify the following key points: (1) the relative impact of truncation error and propagated elevation uncertainty can be captured in a single quality metric, which we coin the Quality Ratio. This metric can be used to compare the accuracy of topographic metrics across DEM spatial resolutions and uncertainty distributions. (2) There exists an optimal grid resolution at which to calculate the topographic metrics slope and aspect for a

given dataset that minimizes the total impact of both truncation error and propagated elevation uncertainty; the distribution of DEM uncertainties leads to spatial variation in the optimal grid resolutions at which to calculate slope and aspect. For the Santa Cruz Island in southern California, we find an optimal grid resolution of four meters, with island-average slope (aspect) errors of 0.25° (0.75°) for truncation error, and 5° (12.5°) for propagated elevation uncertainty. (3) Topographic metrics calculated at sub-optimal grid resolutions will have potentially large errors in their slope and aspect distributions. Resampling grids to the

optimal resolution does not completely ameliorate these issues; it is important to generate the DEM at the optimal resolution from the underlying lidar dataset. (4) Slope and aspect calculations at high DEM resolutions (>3 meters) are significantly impacted by the nonlinear relationship between grid spacing and propagated elevation uncertainty. A DEM with very low uncertainties is required to support robust analysis of fine-scale topography.

Given that grid-resolution-driven effects on regional slope and aspect distributions could have significant impacts on the

interpretation of landscape morphology, we recommend that region-specific optimal DEM resolutions be determined before the calculation of topographic metrics.

*Code and data availability.*  All codes and data for reproducing the results can be found in on GitHub (https://github.com/UP-RS-ESP/TopoMetricUncertainty).

*Author contributions.*  T.S. led the development and writing of the MS, as well as the primary data analysis. A.R. contributed to the develop-

ment of the methods and developed most of the python codes. B.B. contributed to the development of the methods and processed the Lidar data.

*Competing interests.*  The authors declare no competing financial interests.

*Acknowledgements.*  The State of Brandenburg (Germany) through the Ministry of Science and Education and the NEXUS project supported T.S. for part of this study (grant to. B.B.). The authors thank Fiona Clubb and Ben Purinton for comments on an earlier version of the

manuscript.



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
