# Peer review of "Determining the Optimal Grid Resolution for Topographic Analysis on an Airborne Lidar Dataset"

_Earth Surface Dynamics, 2018_

## Referee Comment (RC2) · Anonymous Referee #2 · 18 Feb 2019

The manuscript "Determining the optimal grid resolution..." presents an analysis of two types of errors that affect DEMS and offers a method for finding the DEM resolution that will minimize these errors for real data. I think that this is a very nice study. The analysis is thorough (23 supplemental figures!), and the combination of theoretical derivations, synthetic data, and application to a real dataset makes for a convincing combination. It is definitely something that will be of interest to people working with airborne lidar data, and since you offer a readily applicable method, I think that people will find this quite useful. I find the paper just about ready to publish, so I have only a few minor comments.

I am curious about the impact of irregular topography on the choice of resolution - if you have a stepped landscape (such as terraced hillslopes), would that change the

consideration of the optimal resolution?

You show at the end that resampling gridded data to the optimal resolution is not sufficient to achieve the same quality as creating the DEM at that resolution, but it does seem to improve the slope distributions for the 1m data (it is interesting that resampling the 2 m data appears to move the distribution away from the ideal distribution). Assuming that you have the standard deviations and could calculate the optimal resolution, it seems that resampling could still be helpful if you don't have access to the ungridded point data?

Fig. 7: It's interesting that the optimal aspect spacing for the two highest noise levels is the same. Does this mean that the optimal spacing maxes out at this point?

Fig. 11: I find it hard to see much of anything with the color scale in panel C. Maybe one with more contrast would be better?

Watch your use of grid spacing vs. grid resolution - because higher resolution = smaller grid spacing, it can be a little confusing if you suddenly switch between them, as you do in the caption to Fig. 13.

It would also be nice to keep the figures consistent with slope and aspect shown in the same order (ie. slope always on the left/top and aspect on the right/below) - i.e. in fig. 5 it is reversed from fig. 3.

pg. 22 line 16: I guess you mean <3 meters

---

## Author Comment (AC1)

**Reply to Reviewers – Smith, Rheinwalt, and Bookhagen**

*Reviewer #1 (Remarks to the Author):*

*The manuscript by Smith et al. deals with the identification of the optimal grid resolution for slope and aspect derivation from LiDAR dataset. The topic is relevant and of utmost importance in the quantitative geomorphology and hydrology fields since aspect and, especially, slope are fundamental parameters for different analyses commonly carried out such as synthetic channel network derivation, feature extraction and modeling. The paper presents the theoretical steps and a real-world application of a quality metric for slope and aspect calculations developed to determine the optimal resolution minimizing truncation error and the propagated elevation uncertainty. In my opinion, the work is very well structured and the method is robust and clearly outlined in the manuscript. This innovative approach is highly promising and will be most likely of great interest for the geomorphometric community thanks also to the fact that made available the developed code in the GitHub repository. My only minor concern about this high quality work is related to the assumption that the optimal grid resolution of a Digital Elevation Model (DEM) is the one minimizing the total error. This is certainly true in most cases but for several applications, the choice of the DEM resolution is mainly controlled by the dimension of the feature investigated that is some cases may be quite small (e.g. ephemeral gully, colluvial channels) and, thus, a fine resolution may be the best choice. Typically, this is addressed by choosing a high resolution and then computing geomorphometric parameters using large moving window to filter out the noise. Apart from the sentence at lines 1-2, page 18, this issue is not mentioned in the manuscript. I suggest adding some text on this to better clarify your assumption on the "optimal" grid size.*

Thank you for your comments on our paper. The issue of scale is difficult – there are many studies which use one-meter or sub-meter gridding of airborne lidar (such as we use in this study) to study small-scale features. Our analysis shows that the uncertainties for these very high resolution datasets become large; in many cases the uncertainties can grow larger than the scale of the fine-scale topography that is studied. While such high-resolution analysis should not be ruled out, it is important to be aware of the uncertainties involved, and whether or not the dataset is precise enough to support the analysis of very small landscape features – with or without using larger window sizes as an approach to smooth out noise. We have added an additional comment in the paper (page 18) to further clarify this.

**Specific Comments:**

*L. 22, p. 1 (and throughout the text): lidar-> LiDAR*

In our paper we use the lower-case 'lidar', similar to how radar is used as an abbreviation of RAdio Detection And Ranging. While LiDAR is being used in some literature, the National Academy of Sciences and other national institutions in the UK rely on lidar (https://essp.nasa.gov/essp/files/2018/02/2017-Earth-Science-Decadal-Survey.pdf; http://lidarmag.com/wp-content/uploads/PDF/LiDARNewsMagazine_DeeringStoker-CasingOfLiDAR_Vol4No6.pdf). In the literature, lidar shows the most common usage (http://lidarmag.com/wp-content/uploads/PDF/LiDARNewsMagazine_DeeringStoker-CasingOfLiDAR_Vol4No6.pdf).

*L. 24-25, p.1: please consider providing a range of resolutions for "high" and "low" resolution in place of e.g 1 and 30 meters*

This has been changed in-text.

*L. 3-10, p. 2: I suggest mentioning also stream power studies that are quite relevant in hydrology and soil science.*

We have added some relevant citations here.

*L. 7, p. 8: differently->different?*

This has been corrected.

*Figure 13 caption: sq km->km^2*

This has been changed.

*L. 7, p. 21: also averaging values could be tested as resampling method, maybe the results are different with respect to those obtained using bilinear resampling. In general, I know it is out of the scope of the paper but it would be interesting to compare different resampling techniques in order to suggest the best method to resample resolution*

We have added an additional panel to Figure 15 (also below as Figure 1 of this Reply) which shows the results of nearest-neighbor resampling. The four- and five-meter resampled distributions are significantly worse with nearest-neighbor, but the one- and two-meter resampled distributions match the optimal three-meter distribution quite closely. We attribute this to the methodology by which we created the lidar DEMs – the one-meter data is derived from same set of neighborhoods which are aggregated into the three-meter data. The three-meter data is thus quite similar to a nearest-neighbor resampling of the underlying one-meter DEM. We have added an additional comment on this in the paper.

[Figure]

Figure 1 – Impact of two different resampling mechanisms on slope distributions.

***Reviewer #2 (Remarks to the Author):***

***The manuscript "Determining the optimal grid resolution..." presents an analysis of two types of errors that affect DEMS and offers a method for finding the DEM resolution that will minimize these errors for real data. I think that this is a very nice study. The analysis is thorough (23 supplemental figures!), and the combination of theoretical derivations, synthetic data, and application to a real dataset makes for a convincing combination. It is definitely something that will be of interest to people working with airborne lidar data, and since you offer a readily applicable method, I think that people will find this quite useful. I find the paper just about ready to publish, so I have only a few minor comments.***

***I am curious about the impact of irregular topography on the choice of resolution – if you have a stepped landscape (such as terraced hillslopes), would that change the consideration of the optimal resolution?***

Irregular topography will likely result in two distinct optimal resolutions – in your example the flat and steep parts of the terrain likely have different error distributions, and do not see the same impacts from TE and PEU. If the landscape is analyzed together, the calculated optimal grid resolution might not be applicable to all parts of the landscape, as we found in our analysis of individual watersheds on SCI (MS Figure 13). Depending on the analysis approach, the landscape could be broken into two segments and analyzed separately.

***You show at the end that resampling gridded data to the optimal resolution is not sufficient to achieve the same quality as creating the DEM at that resolution, but it does seem to improve the slope distributions for the 1m data (it is interesting that resampling the 2 m data appears to move the distribution away from the ideal distribution). Assuming that you have the standard deviations and could calculate the optimal resolution, it seems that resampling could still be helpful if you don't have access to the ungridded point data?***

We have added an additional panel to this figure (Figure 15 in the MS, Figure 1 of this Reply), which covers nearest neighbor resampling. Interestingly, the one-meter distribution is almost identical to the three-meter when nearest neighbor resampling is used. We attribute this to the lidar gridding approach we used – the three-meter neighborhood is essentially the same as the nine one-meter neighborhoods which then form the basis for the nearest neighbor resampling. In that specific case, the resampling might improve the error distribution. However, in general, the slope distributions of resampled data will not match the distribution of the target resolution. Any resampling will also introduce new biases into the data which are hard to account for.

***Specific Comments:***

***Fig. 7: It's interesting that the optimal aspect spacing for the two highest noise levels is the same. Does this mean that the optimal spacing maxes out at this point?***

This is due to the limits on grid spacing from our theoretical data – the smallest useful Gaussian Hill we can model is on an 11x11 grid. In theory, the very high noise level case should also have an optimal grid resolution – we just aren't able to model it because the data cannot be gridded any more coarsely. Note that the grid spacing values here are an expression of the width-to-height ratio of the surface, and thus aren't a 'hard' limit on grid spacing, but more related to the dimensions of the grid used to model the Gaussian Hill.

***Fig. 11: I find it hard to see much of anything with the color scale in panel C. Maybe one with more contrast would be better?***

This has been updated.

***Watch your use of grid spacing vs. grid resolution - because higher resolution= smaller grid spacing, it can be a little confusing if you suddenly switch between them, as you do in the caption to Fig. 13.***

Thank you for catching this inconsistency. This has been updated.

***It would also be nice to keep the figures consistent with slope and aspect shown in the same order (ie. slope always on the left/top and aspect on the right/below) - i.e. in fig. 5 it is reversed from fig. 3.***

This has been updated throughout.

***pg. 22 line 16: I guess you mean <3 meters***

This has been fixed.

---

## Referee Report (RR1)

The paper by Smith et al. argues that, for a given airborne lidar dataset, there exists an optimal resolution which minimizes the impact of both gridding errors and any uncertainty in the DEM in the calculation of topographic metrics.

I think this paper should eventually be published but I have serious concerns about the analysis that I urge the authors to consider prior to publication.

1) I am concerned with how the authors created their DEMs. When I create a DEM of a mathematical function I sample the function at regularly spaced points. The resulting DEM is an incomplete representation of the surface but it is an accurate representation where the function is sampled. When I compute slope and aspect using DEMs created this way there are errors associated with discretization, but they are small and converge to zero as the pixel size becomes small.

As such, I was surprised by magnitude and types of errors computed by Smith et al. for their synthetic cases. When I looked more closely, I saw the reason for this discrepancy. If I am interpreting the code https://github.com/UP-RS-ESP/TopoMetricUncertainty/blob/master/gaussian_hill_example.py correctly, Smith et al. have generated their DEMs of Gaussian hills by randomly sampling a Gaussian function and then computing the mean elevation of those random samples within each domain. None of this is explained in the manuscript in the section on synthetic data analysis. Since no information is given, I have no idea how many random samples were used to compute the mean value, or why the authors choose to use the mean value. When scientists grid data from a point cloud, they generally use Independent Distance Weighting (IDW) because this method weighs measurement points close to the sample location more heavily than points father from the sample. The author's approach is not an interpolation of any kind – it treats measurements points far from the DEM grid point location ($x\_i$, $y\_j$) equal to those close to the grid point location. DEM values are supposed to represent the elevation at each point on the surface. DEMs are never supposed to represent the mean elevation within some square domain. Yet, that is how they have been created in this manuscript and I believe that much of the error that the authors are studying is due to the nonstandard way that they have created their DEMs. To address this issue, I urge the authors to explain how their DEMs are created and use IDW to create the DEMs from the point cloud. I would further urge the authors not to assume a random sample in their synthetic DEMs, since lidar data are not a random sample. Before this error is fixed it is difficult to even fully review the paper. However, I will do the best I can, recognizing that this can only be a preliminary review until the DEMs are properly computed.

2) I am having difficulty understanding the error equations. In eqns 4 and 5 there is a partial derivative $\partial$ with another partial derivative $\partial z / \partial x$ as a subscript. I have never seen this syntax before. What does it mean? I am also confused by the reference to epsilons as uncertainties. In the paper, the epsilon values used to compute TE are computed using the standard deviations within each pixel, which is not the same as uncertainty. The uncertainty of a mean value can be quantified using a standard deviation, but only after being divided by the square root of the number of samples used to compute the mean. I am similarly confused by the use of standard deviations without any scaling by the number of samples in the PEU calculations.

3) One of the metrics used by the authors, the truncation error is, according to the authors, "uncertainty associated with the representation of a continuous surface as a grid." However, since landscapes have roughness at all scales (i.e., they are not differentiable and, more broadly, any increase in DEM resolution almost always results in additional real features being resolved in the topography), it is not necessarily the case that a polynomial is a better approximation of the surface than a straight line, as implied by the truncation error and the associated assumption that minimizing TE leads to a better result. I can see how TE would be a useful measure if topography was smooth at small scales, but I don't think this is supported

by observations of actual topography. To address this issue, the authors could explore and defend their choice of TE in landscapes with microtopography present (i.e., nearly all landscapes) or they could perform their analysis without using TE.

4) I don't understand Figure 4. Part B illustrates conceptually how aspect values are pushed away from and towards certain angles. That is not what part A shows. Part A shows that the probability density of 91 degrees is anomalously high and that of 89 degrees is anomalously low. The same bias towards larger values just above angles that are multiples of 45 degrees applies to all other values. I don't understand how this bias occurs but it is certainly not the result of a tendency of the algorithm to result in higher values at angles that are multiples of 45 degrees, as implied by part B.

5) The most common method for determining the appropriate scale for computing slopes and curvatures that reflect landscape-scale attributes is to plot curvature as a function of scale and identify the scaling break following Roering et al., 2010, Evidence for biotic controls on topography and soil production. I think this alternative should be mentioned. At present the paper isn't referenced.

---

## Author Response (AR2)

***Reply to Reviewers – Smith, Rheinwalt, and Bookhagen***

Reviewer #3:

*The paper by Smith et al. argues that, for a given airborne lidar dataset, there exists an optimal resolution which minimizes the impact of both gridding errors and any uncertainty in the DEM in the calculation of topographic metrics.*

*I think this paper should eventually be published but I have serious concerns about the analysis that I urge the authors to consider prior to publication.*

> Thank you for your detailed comments, thorough review, and time spent with the manuscript and code. We have done our best to respond to each of the comments individually, and make appropriate changes to the MS.

Specific Comments:

*1) I am concerned with how the authors created their DEMs. When I create a DEM of a mathematical function I sample the function at regularly spaced points. The resulting DEM is an incomplete representation of the surface but it is an accurate representation where the function is sampled. When I compute slope and aspect using DEMs created this way there are errors associated with discretization, but they are small and converge to zero as the pixel size becomes small. As such, I was surprised by magnitude and types of errors computed by Smith et al. for their synthetic cases.*

*When I looked more closely, I saw the reason for this discrepancy. If I am interpreting the code* *https://github.com/UP-RS-ESP/TopoMetricUncertainty/blob/master/gaussian_hill_example.py* *correctly, Smith et al. have generated their DEMs of Gaussian hills by randomly sampling a Gaussian function and then computing the mean elevation of those random samples within each domain. None of this is explained in the manuscript in the section on synthetic data analysis. Since no information is given, I have no idea how many random samples were used to compute the mean value, or why the authors choose to use the mean value. When scientists grid data from a point cloud, they generally use Independent Distance Weighting (IDW) because this method weighs measurement points close to the sample location more heavily than points father from the sample. The author's approach is not an interpolation of any kind – it treats measurements points far from the DEM grid point location (x_i, y_j) equal to those close to the grid point location. DEM values are supposed to represent the elevation at each point on the surface. DEMs are never supposed to represent the mean elevation within some square domain. Yet, that is how they have been created in this manuscript and I believe that much of the error that the authors are studying is due to the nonstandard way that they have created their DEMs. To address this issue, I urge the authors to explain how their DEMs are created and use IDW to create the DEMs from the point cloud. I would further urge the authors not to assume a random sample in their synthetic DEMs, since lidar data are not a random sample. Before this error is fixed it is difficult to even fully review the paper. However, I will do the best I can, recognizing that this can only be a preliminary review until the DEMs are properly computed.*

> We believe that there has been a substantial misunderstanding of our methods here, and we would like to clarify that we do not randomly sample a Gaussian function for our

synthetic data analysis, but rather sample a mathematical function at evenly spaced points, as the reviewer suggested is best practice.

In our github code, we provide two forms of synthetic data, one which is representative of a point cloud, and one which is created natively on a grid using equal-spaced sampling as expected by the reviewer. The functions found here: https://github.com/UP-RS-ESP/TopoMetricUncertainty/blob/master/surfaces.py are used to create our synthetic data that are used in the first part of the MS. We do not create a point cloud and take a mean of randomly-sampled points, as the reviewer suggests, but rather calculate the mathematical value of the given exponential term in the Gaussian function at each sampled (x, y) coordinate. This is explained in the MS on page 4, line 10, although we agree that this could be made more clear by explaining our gridding procedure or linking to the specific code used to create the surfaces. The secondary set of code that the reviewer linked to is a toy example with synthetic lidar data, and is not used in the MS. During data analysis and manuscript preparation we explored the impact of randomly sampling points akin to a lidar dataset. We were especially interested in the impact of X-Y offsets (gridding data with an uncertainty in the X/Y location) and point clustering. We do not present these results in the manuscript (as they would extent the already long manuscript even more), but kept this part of the code on the github archive. We have updated the Github landing page to be more explicit as to what each piece of code does, and which are used in the MS.

Given that we simply sample our Gaussian function at each point on our grid in equal steps, we can also sample the slope and aspect of that function (see Equations 1, 2), which are well-known at each point of our grid for a differentiable function. We then use those 'perfect' samples of slope and aspect at each sample point to compare to the slope and aspect grids calculated on our 'perfect' elevation grid using various algorithms. When we compare the computed difference of our 4-neighbor method to the derived values of TE on the same ideal grid (compare MS Figure 3 and Supplemental Figure 2), the magnitudes and directions are the same. We thus conclude that our TE model accurately captures the magnitude of algorithm-induced error on the synthetic surface.

We also believe there was also a misunderstanding of our methods with concern to the generation of the DEMs from the lidar point cloud. To clarify, we have generated our lidar DEMs using a triangular irregular network (TIN) approach, which is also widely used in the lidar community and implemented by LASTools (e.g., Isenburg et al, 2006). This is an alternative method to the weighted mean provided by IDW techniques, and also results in reliable lidar DEMs. To supplement our analysis and address the concerns related to DEM generation, we have generated DEMs from the same source lidar data using several alternative techniques, including IDW with several different parameters (Table 1 of this reply, new Supplementary Table 1, and Supplementary Figures 13-21). The difference between these elevation models is very small (see Figure 1 of this Reply and Supplementary Figures 13-21). For detailed processing methodologies, we refer to our manual here: https://github.com/BodoBookhagen/Lidar_PC_interpolation

**Table S1.** Listing of interpolation schemes used to create DEMs for SCI. For a more in-depth discussion of each method, we refer to our github page: https://github.com/UP-RS-ESP/TopoMetricUncertainty.

| Scheme Name | Software/Algorithm Basis | Figure |
|---|---|---|
| Triangular Irregular Network | blast2dem, LAStools, LAStools (2017) | |
| Triangular Irregular Network, Delauney Method | triangulation, GMT, Watson (1982) | Figure S13 |
| Local Mean | blockmean, GMT, Wessel and Luis (2017) | Figure S14 |
| Local Median | blockmedian, GMT, Wessel and Luis (2017) | Figure S15 |
| Inverse Distance Weights, power=1, radius=grid size $\times \frac{\sqrt{2}}{2}$ | gdal_grid, GDAL (gdal.org), Shepard (1968) | Figure S16 |
| Inverse Distance Weights, power=1, radius=grid size $\times \sqrt{2}$ | gdal_grid, GDAL (gdal.org), Shepard (1968) | Figure S17 |
| Inverse Distance Weights, power=1, radius=grid size $\times 2\sqrt{2}$ | gdal_grid, GDAL (gdal.org), Shepard (1968) | Figure S18 |
| Inverse Distance Weights, power=2, radius=grid size $\times \frac{\sqrt{2}}{2}$ | gdal_grid, GDAL (gdal.org), Shepard (1968) | Figure S19 |
| Inverse Distance Weights, power=3, radius=grid size $\times \frac{\sqrt{2}}{2}$ | gdal_grid, GDAL (gdal.org), Shepard (1968) | Figure S20 |
| Inverse Distance Weights, power=2, radius=grid size $\times \frac{\sqrt{2}}{2}$ | Points2Grid, pdal (pdal.io), Shepard (1968) | Figure S21 |

Table 1 – List of lidar interpolation schemes used in the preparation of this manuscript. We use the TIN method from LAStools as our primary data source, but have also examined DEMs created with each other method. We refer to Supplementary Figures 13-21 for detailed comparisons of Elevation, Slope, TE, and PEU for each DEM generation method. One figure is also presented below in Figure 1 of this reply.

[Figure]

Figure 1 – Impact of changed DEM interpolation method on a 1m DEM. Left column shows difference for a zoomed in region of SCI for (A) elevation, (C) slope, (E) TE, and (G) PEU when a simple inverse distance weighting scheme is used as compared to the TIN method discussed in the MS. Color bar scaled from 5[th] to 95[th] percentiles. Right column shows the density histogram of the differences for the whole Pozo catchment. Red lines represent the 25[th] and 75[th] percentiles, with a black line centered on zero. Histograms scaled between 5[th] and 95[th] percentiles. While elevation differences are relatively small, they engender much larger slope errors. The difference in calculated TE and PEU between methods remains small.

Regardless of what gridding method is used, there will be some uncertainty in the derived elevation grid. This uncertainty is modified by the terrain type, accuracy of the lidar data, sampling scheme, and many other factors (as mentioned in the MS, page 12, Line 14). While we agree that we do not have a perfect error model – we assume that the standard deviation of the community of points in each lidar grid cell search radius is representative of the uncertainty in that lidar point elevation – we argue that the TE and PEU effects we see are independent of the error model used. We have experimented with different error models, including preferentially selecting more reliable points with lower scan angles (Rheinwalt and Bookhagen, 2018), but the spatial distribution of errors remains similar. Importantly, if a different error model exists, it can be directly included in our approach. Elevation uncertainty will be propagated into the slope and aspect calculations regardless of the DEM used, and the effect of DEM uncertainty on slope and aspect calculations will increase as the spacing of the DEM approaches the uncertainty magnitude.

*2) I am having difficulty understanding the error equations. In eqns 4 and 5 there is a partial derivative $\partial_\iota$ with another partial derivative $\partial_\iota$ $/\partial_\iota$ as a subscript. I have never seen this syntax before. What does it mean? I am also confused by the reference to epsilons as uncertainties. In the paper, the epsilon values used to compute TE are computed using the standard deviations within each pixel, which is not the same as uncertainty. The uncertainty of a mean value can be quantified using a standard deviation, but only after being divided by the square root of the number of samples used to compute the mean. I am similarly confused by the use of standard deviations without any scaling by the number of samples in the PEU calculations.*

We have reviewed the equations here, and want to clarify our notation. We used a shorthand, which is more common in the physics community, to avoid writing double partial derivatives. We quantify the change in topographic metrics with a change of the topography – essentially dS / d(dz/dx), which was not clearly stated. We have reformatted each of the equations in the updated MS to use the classic notation.

In reference to the second point, we think that the reviewer misunderstood our DEM generation method. As we mentioned above in the reply to the first comment, we use equally spaced samples ('gridding') of the mathematical function to create our elevation metrics, so we do not have any uncertainty or standard deviation to include in our TE calculation. We have explored these effects, but did not report them in the manuscript, as they require further analysis and are more appropriate for a subsequent paper.

The TE, as represented by Equations 4/5, does not involve the standard deviation, as it is based simply on the finite difference operation performed on the gridded dataset. The epsilon metric, defined in Equation 3, is an inherent error due to the second order finite difference approximation used here to calculate slope and aspect, and does not take into account any uncertainty in the elevation measurements. We only introduce a standard deviation in Equations 10/11, where the PEU is calculated independently of the TE.

We are unsure as to the reviewer's last point "*I am similarly confused by the use of standard deviations without any scaling by the number of samples in the PEU calculations*". The standard deviation is always calculated with knowledge of the number of samples – the formula for STD is the square root of the sum of squared differences from the mean divided by one minus the sample size. Perhaps this confusion is related

to the misunderstanding of our synthetic data creation methods. In any case, we calculate our PEU formulas by propagating the standard deviation of the elevation data into the directional derivatives used to calculate slope and aspect to arrive at a value for PEU.

*3) One of the metrics used by the authors, the truncation error is, according to the authors, "uncertainty associated with the representation of a continuous surface as a grid." However, since landscapes have roughness at all scales (i.e., they are not differentiable and, more broadly, any increase in DEM resolution almost always results in additional real features being resolved in the topography), it is not necessarily the case that a polynomial is a better approximation of the surface than a straight line, as implied by the truncation error and the associated assumption that minimizing TE leads to a better result. I can see how TE would be a useful measure if topography was smooth at small scales, but I don't think this is supported by observations of actual topography. To address this issue, the authors could explore and defend their choice of TE in landscapes with microtopography present (i.e., nearly all landscapes) or they could perform their analysis without using TE.*

Our metric of TE is technique based – when a three-point difference method is used to calculate a terrain derivative, there is always some error, excepting when the surface is perfectly flat. If we assume that additional features will always be resolved at higher resolutions, for example by modelling terrain as a fractal surface, slope is no longer defined, because the surface is not differentiable. In the case of our synthetic data (the Gaussian Hill), or with the gridded lidar data, we assume a differentiable surface, in which case slope is defined, and we can then calculate the magnitude of the TE for our finite-difference method. We can further test the magnitude of our TE on our synthetic surface, as described on page 7 of the MS, and check that the magnitude of TE from the finite-difference slope analysis matches the analytical solution for TE (compare Figures 3 and Supplemental Figure 2).

In our analysis, TE will always decrease with increasing spatial resolution – a finer grid will have less and less truncation error, whether or not the underlying surface is smooth or rough. However, depending on the application, one may not always want to minimize the TE (ie, to choose only the finest available spatial resolution) as PEU has an increasing effect at very fine scales. At the scale of microtopography, the uncertainty in the elevation data is often as large as the microtopography being resolved – for example, a DEM with a vertical accuracy of 10cm (the accuracy of a very good airborne lidar dataset) can resolve 10cm high features, but cannot decide if those features are signal or noise. The effect is even more pronounced when that DEM is then used to calculate slope or aspect (see Figure 1 of this Reply, where small elevation differences lead to large slope differences). At the fine scale, the PEU for slope and aspect grows very quickly, independent of the shrinking TE. As such, we argue that there exists a scale where the two independent errors and uncertainties of TE and PEU are co-minimized and give slope and aspect estimations with the highest statistical reliability.

*4) I don't understand Figure 4. Part B illustrates conceptually how aspect values are pushed away from and towards certain angles. That is not what part A shows. Part A shows that the probability density of 91 degrees is anomalously high and that of 89 degrees is anomalously low. The same bias towards larger values just above angles that are multiples of 45 degrees applies to all other values. I don't understand how this bias occurs but it is certainly not the result of a tendency of the algorithm to result in higher values at angles that are multiples of 45 degrees, as implied by part B.*

We are confident in the conceptual diagram shown in Part B, which is based on the magnitudes of TE shown in Figure 3. There exist regions of convergence (e.g., 45 degrees is bounded by red (+) on the left and blue (-) on the right, so values are pushed towards it) and divergence (e.g., 90 degrees). Furthermore, the magnitude of those biases is modified by the slope of the surface – there are larger aspect biases in flat areas (see Figure 3).

However, the reviewer is correct in noting that Figure 4A does not follow directly to Figure 4B – this was an oversight in the MS that we have fixed. In essence, the 'spikes' seen in Figure 4A show the impact of the choice of aspect bin centers, not the convergence/divergence shown in Figure 4B. As the input aspect data is on a square grid, there are naturally more pixels at the cardinal directions. As the histogram function uses half-open bins – e.g., [44,45) and [45,46) – there are significantly fewer points just below the cardinal directions (and other multiples of 45, such as 22.5, etc), and significantly more in the bin above. At the scale of our synthetic data (1001x1001 sized grid), this translates to 1-2 extra pixels in the cardinal direction bins, which create those spikes in the frequency distribution. The impact of switching from bins starting at 0 or 0.5 can be seen in Figure 2 of this reply.

[Figure]

Figure 2 – Impact of bin centers. Black lines use bins centered over integer numbers (e.g., [44.5, 45.5), [45.5, 46.5)). Red lines use bins centered over halves (e.g., [44, 45), [45, 46)). While both binnings result in spikes in the aspect distributions, the very large spikes at the cardinal directions are more pronounced in the centered (red) binning.

Unfortunately, there does not exist a binning which can suppress all of the spikes in the aspect distribution of a square grid. Referring back to our original Figure 4, we only expect the effect shown in Figure 4B to manifest with very fine binnings (on the order of the error magnitudes, 1e-4), which will only be seen in very large datasets, and should not have an impact on our aspect bins at the scale of our synthetic data grid spacing. For DEMs with larger truncation errors, for example a DEM with a 10-m grid-cell size, we would, however, expect the TE to shift pixels in and out of certain aspect bins, as is suggested by Figure 3 and Figure 4B.

In light of this clarification, we have modified Figure 1, which is also influenced by the choice of bin centers, and removed Figure 4 so as not to cause confusion. We have also updated text in several places in the MS where the binning effect was misrepresented as a TE impact.

*5) The most common method for determining the appropriate scale for computing slopes and curvatures that reflect landscape-scale attributes is to plot curvature as a function of scale and identify the scaling break following Roering et al., 2010, Evidence for biotic controls on topography and soil production. I think this alternative should be mentioned. At present the paper isn't referenced.*

> We have added this citation as an alternative method of choosing the appropriate scale for DEM analysis (Page 18, Line 10).

[revised manuscript text omitted]